# Prediagnostic loss to follow-up in an active case finding tuberculosis programme: a mixed-methods study from rural Bihar, India

Tushar Garg [1], Vivek Gupta [2], Dyuti Sen [3], Madhur Verma [4], Miranda Brouwer [5], Rajeshwar Mishra,[1,6] Manish Bhardwaj [3]

RM and MaB are joint senior authors.

For numbered affiliations see end of article.

**Correspondence to**
Dr Tushar Garg;
tgarg@innovatorsinhealth.org

## ABSTRACT

**Objective** To quantify the prediagnostic loss to follow-up (PDLFU) in an active case finding tuberculosis (TB) programme and identify the barriers and enablers in undergoing diagnostic evaluation.

**Design** Explanatory mixed-methods design.

**Setting** A rural population of 1.02 million in the Samastipur district of Bihar, India.

**Participants** Based on their knowledge of health status of families, community health workers or CHWs (called accredited social health activist or locally) and informal providers referred people to the programme. The field coordinators (FCs) in the programme screened the referrals for TB symptoms to identify presumptive TB cases. CHWs accompanied the presumptive TB patients to free diagnostic evaluation, and a transport allowance was given to the patients. Thereafter, CHWs initiated and supported the treatment of confirmed cases. We included 13 395 community referrals received between January and December 2018. To understand the reasons of the PDLFU, we conducted in-depth interviews with patients who were evaluated (n=3), patients who were not evaluated (n=4) and focus group discussions with the CHWs (n=2) and FCs (n=1).

**Outcome measures** Proportion and characteristics of PDLFU and association of demographic and symptom characteristics with diagnostic evaluation.

**Results** A total of 11 146 presumptive TB cases were identified between January and December 2018, out of which 4912 (44.1%) underwent diagnostic evaluation. In addition to the free TB services in the public sector, the key enablers were CHW accompaniment and support. The major barriers identified were misinformation and stigma, deficient family and health provider support, transport challenges and poor services in the public health system.

**Conclusion** Finding the missing cases will require patient-centric diagnostic services and urgent reform in the health system. A community-oriented intervention focusing on stigma, misinformation and patient support will be critical to its success.

## INTRODUCTION

The End TB strategy targets an 80% reduction in tuberculosis (TB) incidence by 2030.[1] However, as many as 4.1 million (39%)

### Strengths and limitations of this study

► First such study to explore the reasons for prediagnostic loss to follow-up in a tuberculosis programme.
► A mixed-methods design that includes the views of both patients and community health worker.
► The study used operational data from a routine programmatic setting.
► Since intervention removed some barriers, all findings are not necessarily generalisable.
► No record of the actual number of people screened before being referred to the programme.

patients globally are not notified, indicating a mix of under-reporting and underdiagnosis.[2] With an incidence rate of 199 per 100 000 population in 2018, India alone has about a million such 'missing' cases.[3]

Most patients with TB are identified through passive case finding where patients reach out to a provider.[4] However, it leads to delays in diagnosis and loss to follow-up (LFU) during diagnostic and treatment phases.[5] In fact, the Indian public sector cascade estimates a 28% LFU at test access among incident cases.[6] The major challenges in diagnosis are poor geographical and financial access to healthcare and failure to test when people do present at the facility.[7] Active case finding (ACF) can solve some of these challenges by proactively taking health services to the people in the community.[2 8]

Diverse ACF strategies have demonstrated increased TB case detection with early diagnosis.[4 9–11] Nonetheless, multiple ACF interventions report a high prediagnostic loss to follow-up (PDLFU). An Indian study reported only 22% of the people with presumptive TB reaching a microscopy centre, while a study in Myanmar estimated only 51.4% of patients with abnormal chest X-ray (CXR) getting their sputum examined.[12 13] Although the

quantum of the PDLFU is known, its underlying causes are yet to be investigated.

We studied the PDLFU in a community-based ACF programme implemented in India in 2018. The objectives were to assess proportion and characteristics of PDLFU cases, identify the risk factors and understand the reasons for the PDLFU, including the barriers and enablers in accessing diagnostic evaluation.

## METHODS
### Study design
We used an explanatory mixed-methods study design, where the quantitative phase (cohort analysis) was followed by a qualitative phase (descriptive design).[14]

### Study setting
#### General setting
The study was conducted in the Sarairanjan, Bibhutipur and Ujiarpur blocks (equivalent to TB unit or TU) of Samastipur district in Bihar with a combined population of 1 021 483. The highest earning member in 69.8% households earned less than INR 60 000 annually (~US$852), and 63.5% of the population was literate.[15] Female literacy rate was 20% lower than male, and infant mortality rate per 1000 live births was 60 for females against that of 48 for males.[16] In 2017, the public TB case notification rate in the district was 55 per 100 000 population with a pretreatment LFU of 25%.[17] Furthermore, patients preferred private sector to access TB care.[18]

The population was serviced by 12 primary health centres (PHCs) in the public health system (PHS).[19] Within the Revised National TB Control Program (RNTCP), four designated microscopy centres (DMCs) run by laboratory technicians provided sputum microscopy. A senior treatment supervisor managed each TU and a senior treatment laboratory supervisor managed a group of DMCs.

The TB care in the community were provided by accredited social health activist (ASHA). ASHA are a cadre of community health worker (CHW) assigned for every 1000 population within the National Health Mission (NHM). Her role is to be a facilitator, mobiliser and community service provider for various public health programmes.[20] ASHA receives an activity-based remuneration of INR 1000–1500 (~US$ 14–21) for drug-sensitive TB treatment support and INR 5000 (~US$ 72) for drug-resistant treatment support.[21]

Anganwadi workers (AWW) are a CHW in the Integrated Child Development Services programme and receive a fixed monthly honorarium. Through Angadwadi Centres, they provide early childhood care and education for children up to 6 years. However, they are directly not involved in TB care.[22]

#### Specific setting
Supported by TB REACH, we implemented a community-based ACF project in collaboration with RNTCP and NHM. ACF was added to the routine RNTCP programme with interventions in the community as well as the PHS. Its key components were community referral, symptom-based screening at patient's home, transport allowance to patients, free diagnostic evaluation and diagnosis and treatment assisted by ASHAs. There were no consistent ACF campaigns prior to this programme in our study population.

We engaged various health workers, including ASHA, AWW and registered medical practitioner (informal providers known as RMP). A TB awareness meeting was organised in the community to kickstart the programme. The health workers and laypersons were asked to refer people who may have TB to the programme. Since they are aware of the health status of families in their community, they were well-placed to identify such people in their routine work. These community referrals were reported to a field coordinator (FC) of the project, who screened them at their home for TB symptoms. The presumptive TB cases were those with one or more of the following symptoms: cough of ≥2 weeks, sputum in cough, haemoptysis in last 6 months, chest pain in last 1 month, fever of ≥2 weeks, night sweats for ≥2 weeks, severe weight loss in the last 3 months and swelling in a lymph node.

The diagnostic algorithm in the study followed RNTCP's recommendations.[21] All presumptive TB cases were tested using sputum microscopy and CXR. Patients with a positive smear, or those with a negative smear but abnormal CXR, or patients with a negative smear but clinically suspect were offered a GeneXpert test using sputum collection and transport mechanism. ASHAs assisted presumptive TB patients in reaching the diagnostic centres and accompanied them through the diagnostic process. Patients received a transport allowance and diagnostic tests were free of cost. If the PHS physician confirmed TB diagnosis, treatment was initiated under ASHA's supervision. ASHA delivered drugs to the patient, ensured follow-up tests and treatment adherence and monitored for adverse effects (figure 1).

For each confirmed TB case, a conditional incentive of INR 200 (~US$3) for community referral and INR 300 (~US$5) for diagnostic support was given to the CHWs. These incentives to the CHW were in addition to the RNTCP's treatment support incentives. The project's FCs received a fixed monthly salary.

The project used a Management Information System (MIS) developed in Microsoft Excel 2016, which maintained individual patient data electronically. On receipt of a community referral, the FC enlisted them in a referral register with a unique ID, and a separate form was completed for screening. The status of each community referral was tracked using the referral register. For presumptive TB cases undergoing diagnostic evaluation, an individual folder with their forms and diagnostic reports was maintained. Data were entered weekly into the MIS, and at least 20% entries were verified monthly in a two-person formation to assess data entry quality.

| Term | Definition |
|---|---|
| Community referral | A person at risk of TB referred for TB screening to the programme by a CHW (ASHA or AWW) or an informal provider or community during their routine work |
| Presumptive TB patient | A person identified with one or more of the following symptoms of TB in the screening process at their home: cough ≥ 2 weeks; sputum in cough, haemoptysis in last 6 months; chest pain in last 1 month; fever ≥ 2 weeks; night sweats ≥ 2 weeks; severe weight loss in last 3 months; swelling in a lymph node |
| Diagnostic evaluation | A presumptive TB patient tested using one or more of a sputum microscopy or CXR or GeneXpert or extrapulmonary TB testing (USG or FNAC) within 30 days of screening |
| Pre-diagnostic loss to follow-up | A presumptive TB patient not undergoing a diagnostic evaluation |

**Figure 1** Operational definitions used in the programme. ASHA, accredited social health activist; AWW, Anganwadi worker; CHW, community health worker; CXR, chest X-ray; FNAC, fine-needle aspiration cytology; TB, tuberculosis; USG, ultrasonography.

### Study population and study period
#### Quantitative
All community referrals between 1 January 2018 and 31 December 2018 in the three selected blocks of Samastipur district were included in the study. There was no sampling involved.

#### Qualitative
Presumptive TB cases referred for diagnostic evaluation were interviewed. In-depth interviews were conducted with three presumptive TB cases who were evaluated (two women, one man, age range: 17–30 years) and four presumptive cases who were not evaluated (two women, two men, age range: 6–65 years). A purposive sample of ASHAs and FCs was selected for the focus group discussions (FGDs). Three FGDs were conducted with a group of eight ASHAs (all women, age range: 24–49 years), 11 ASHAs (all women, age range: 22–54 years) and 9 FCs (six women, three men, age range: 20–38 years). The average duration was 36 min (range 23–51 min). A diverse sample of men and women across age groups was purposively selected for diversity and interviewed until saturation was achieved.

### Data variables, sources of data and data collection
#### Quantitative
Data on the community referrals' characteristics (location, referral source, age and gender), screening criteria (date of screening, symptoms and outcome) and diagnostic evaluation (date of test and type of test) were extracted from the MIS into a structured proforma. The data extracted from the MIS was deidentified before its export for analysis (figure 1).

#### Qualitative
Presumptive TB cases were interviewed by TG (a male medical doctor trained in qualitative research) and DS (a female economist trained in qualitative research), and FGDs were conducted by TG and RM (a male professor of social psychology). All the three investigators have adequate experience of the sociocultural context of the region and understanding of TB programmes. After obtaining a participant's consent, the interviews were conducted at a time and place convenient for the participant, and their privacy was ensured. The regional vernacular was used for interactions, and it was recorded after obtaining consent. The objectives of the study were explained to the participants. No participant denied permission for interview, and there were no repeat interviews. A topic guide was used for FGDs and an interview guide for in-depth interviews to explore the enabling factors and barriers for diagnostic evaluation. Appropriate probes were used for clarity and to elicit information. The information was debriefed for participant validation after the interview, but the transcripts were not returned to them.

### Analysis and statistics
#### Quantitative
The data were analysed using EpiData Analysis (V.2.2.2.183, EpiData Association, Odense, Denmark) and Stata (V.15.1, StataCorp LLC, College Station, Texas, USA). Patients not undergoing a diagnostic test for TB within 30 days of referral were considered PDLFU. Patients with diagnostic test after 30 days (n=140) were excluded from the diagnostic evaluation category. In the event that patients had a diagnostic test performed prior

to community referral, the time to diagnostic visit was set as 0 days (n=486). Age was missing for 242 community referrals. As applicable, variables were summarised with mean (and SD or SD) or median (and IQR) based on statistical distribution of data, or frequencies and percentages. Association of demographic and symptom characteristics with diagnostic evaluation was analysed using $\chi^2$ test and unadjusted relative risk (RR) with 95% CI was calculated.

A multivariate analysis using Poisson's regression was used to calculate adjusted relative risk (aRR) with 95% CI. Collinearity was assessed using variance inflation factors during model building. Likelihood ratio tests were used to identify factors that contributed significantly to the model. P value ≤0.05 was considered statistically significant. A sensitivity analysis (univariate and multivariate) was performed by excluding patients whose diagnostic test was performed prior to screening (n=486) and also by classifying patients who underwent diagnostic evaluation after 30 days as non-PDLFU (n=140).

### Qualitative

The transcripts were prepared on the same day using audio recording and field notes by TG. Manual descriptive content analysis was performed by two independent, trained researchers (TG and DS) to generate categories and themes.[23] Any discrepancies between the two were resolved through discussion. These were discussed and reviewed by RM to avoid subjective bias. The codes and themes were related back to the original data to ensure that the results reflect the data.[24]

### Patient and public involvement

Patients were neither involved in the study design nor in the interpretation of patient relevant outcomes. Nonetheless, patient's views were sought in the qualitative interviews and included in the results. The results of this study will be communicated to the patients and the public through a vernacular newsletter.

### RESULTS
### Care cascade and characteristics of presumptive TB cases

We received a total of 13 395 community referrals, out of which 90.9% (n=12 180) were screened for symptoms. Of those screened, 91.5% (n=11 146) were presumptive TB cases, and referred for diagnostic evaluation (figure 2).

There was nearly equal representation of presumptive TB cases from all the three blocks, and ASHAs identified most of them (75.6%). The mean age of presumptive TB cases was 35 years with majority in the 15–44 years age group (41.8%). There were more men (52.2%) than women (table 1).

The most common symptoms among presumptive TB cases were cough of ≥2 weeks (79.8%), severe weight loss in the last 3 months (74.6%), fever of ≥2 weeks (73.9%) and chest pain in the last 1 month (63.4%), while haemoptysis in the last 6 months (12.1%) was the least common.

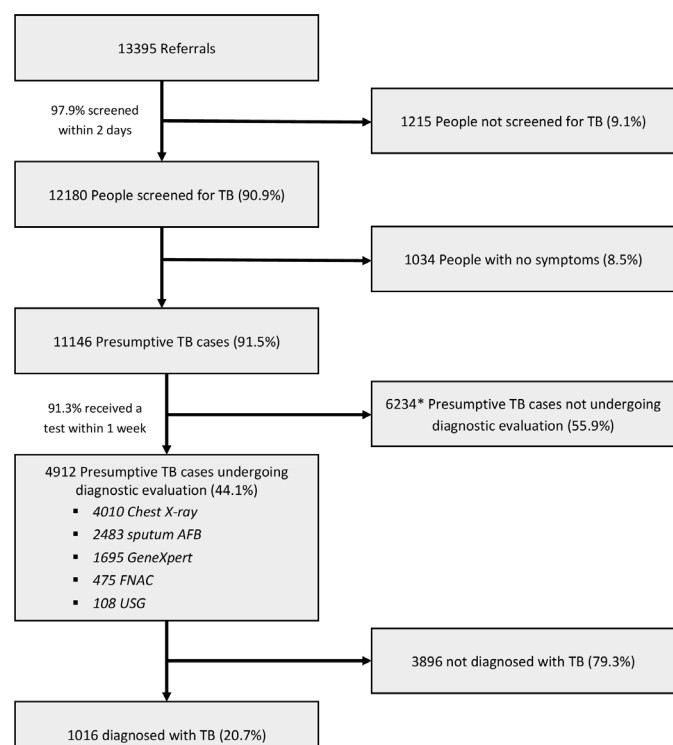

**Figure 2** TB care cascade between January 2018 and December 2018 from community referral to diagnostic evaluation in a community-based ACF programme in Samastipur, India. All percentages are calculated as a proportion of the number of participants entering the previous step of the cascade. *Includes 140 presumptive TB cases undergoing diagnostic evaluation beyond 30 days from screening. ACF, active case finding; FNAC, fine-needle aspiration cytology, TB, tuberculosis; USG, ultrasonography.

Nearly one-fifth of the presumptive TB cases reported a previous history of anti-TB treatment (22.4%), and 20.5% were tobacco users (table 2).

### Characteristics of prediagnostic LFU and associated risk factors

Nearly 44% (n=4912) of the presumptive TB cases received diagnostic evaluation. The prediagnostic LFU was highest among younger (<15 years) presumptive TB cases (68.1%), while lesser among those who were referred by an RMP (43.9%) and complained of haemoptysis in the last 6 months (36.7%). The median time to diagnosis was 1 day (IQR=3).

On multivariate analysis, presumptive TB cases who were <15 years of age (aRR=1.2, 95% CI 1.2 to 1.3, p=0.00) were more likely to be PDLFU. Previous history of TB treatment (aRR=0.7, 95% CI 0.7 to 0.8, p=0.00) and haemoptysis in last 6 months (aRR=0.7, 95% CI 0.6 to 0.7, p=0.00) decreased the chances of PDLFU (table 2).

In the sensitivity analysis, no change was observed in association of the risk factors with PDLFU on either excluding patients who were diagnosed prior to screening or classifying patients who underwent diagnostic evaluation after 30 days as non-PDLFU (online supplementary table 1).

Table 1  Demographic profile of referrals, screened cases and presumptive TB cases identified between January 2018 and December 2018 in Samastipur, India

| Characteristics | Community referrals, n (%) | | Screened for TB, n (%) | | Presumptive TB cases, n (%) | |
|---|---|---|---|---|---|---|
| Total | 13 395 | | 12 180 | | 11 146 | |
| Block | | | | | | |
| Ujiarpur | 4275 | (31.9) | 3926 | (32.2) | 3620 | (32.5) |
| Bibhutipur | 4609 | (34.4) | 3994 | (32.8) | 3673 | (33) |
| Sarairanjan | 4511 | (33.7) | 4260 | (35.0) | 3853 | (34.5) |
| Age (years) | | | | | | |
| <15 | 2847 | (21.3) | 2611 | (21.4) | 2385 | (21.4) |
| 15–44 | 5641 | (42.1) | 5147 | (42.3) | 4661 | (41.8) |
| 45–64 | 3330 | (24.9) | 3032 | (24.9) | 2789 | (25.0) |
| ≥65 | 1335 | (10.0) | 1243 | (10.2) | 1169 | (10.5) |
| Missing | 242 | (1.8) | 147 | (1.2) | 142 | (1.3) |
| Gender | | | | | | |
| Male | 6973 | (52.1) | 6391 | (52.5) | 5827 | (52.2) |
| Female | 6422 | (47.9) | 5789 | (47.5) | 5319 | (47.7) |
| Source of referral | | | | | | |
| ASHA | 10091 | (75.3) | 9210 | (75.6) | 8428 | (75.6) |
| AWW | 105 | (0.8) | 101 | (0.8) | 88 | (0.8) |
| RMP | 724 | (5.4) | 661 | (5.4) | 617 | (5.5) |
| Community | 2475 | (18.5) | 2208 | (18.1) | 2013 | (18.1) |

ASHA, accredited social health activist; AWW, Anganwadi worker; RMP, registered medical practitioner; TB, tuberculosis.

## Enablers to access the first diagnostic evaluation

In the interviews, patients reported the following enablers for diagnostic evaluation: transport allowance for travel to the hospital, free services in the PHS and knowledge of the PHS procedures, accompaniment of ASHA and her assistance in the diagnostic process, and ASHA's understanding of the PHS functioning. (figure 3)

I started facing financial problems after going to the private hospital… Had I known the quality of care in the government hospital earlier, I would have come here. It helped that I didn't have to spend any money. I went by a vehicle to the PHC and consulted the doctor for free of cost. (23 years old male patient)

I didn't have to go to the hospital after submitting the sample for diagnostics… ASHA took me to the hospital and now sends the medicine… I was bony and couldn't even move a step earlier… I became better only from these government drugs; rest were useless. Between RMP and private clinic, we spent nearly ₹20 000 (~USD 292). (17-year-old female TB patient on medication)

When I had chest pain, ASHA told me that we'll have to go to the PHC, tests will be free, and I'll get return fare from my home. I thought I'll get better and decided to go. I got a blood test and x-ray done. Reports were normal and I got some medicines. I felt better… If you know the hospital and workers there, it is easier

to get things done. Else, you have to spend a lot of time. (30-year-old female patient)

The ASHAs and the FCs expressed transport allowance, incentive to ASHA, positive patient experience and patient accompaniment as enablers (figure 3).

If one patient becomes better, she also advises others. Even if we don't know, the patient who becomes better asks their ill neighbor to call the ASHA for everything. Our mobile number is available in every household. (ASHA)

Sometimes, patients do not have any shared conveyance from their home. In such cases, patients are unable to come, and transport allowance or hiring an autorickshaw for, say 10 patients of the village, really helps. That way, they get consultation and lab tests at PHC, then chest x-ray at another diagnostic facility, and get dropped at their home. (Field coordinator)

ASHA's income is from incentives, and if she gets incentives in time, she takes up those activities. (Field coordinator)

## Barriers to access the first diagnostic evaluation

Barriers were coded from the interview and FGD transcripts into 5 categories and 27 codes. These barriers are listed in the figure 3 and described below.

**Table 2** Characteristics of and risk factor for prediagnostic loss to follow-up (PDLFU) among presumptive TB cases referred for diagnostic evaluation between January 2018 and December 2018 in Samastipur, India

| Characteristics | Presumptive TB cases, n | Not evaluated, n (%) | | Unadjusted RR (95% CI) | | P value | Adjusted RR* (95% CI) | | P value |
|---|---|---|---|---|---|---|---|---|---|
| Total | 11146 | 6234 | (55.9) | | | | | | |
| Block | | | | | | | | | |
| Ujiarpur | 3620 | 2060 | (56.9) | 1.0 | (1.0 to 1.1) | 0.022 | – | | |
| Bibhutipur | 3673 | 2083 | (56.7) | 1.0 | (1.0 to 1.1) | 0.033 | – | | |
| Sarairanjan | 3853 | 2091 | (54.3) | | Ref | | – | | |
| Age (years) | | | | | | | | | |
| <15 | 2385 | 1625 | (68.1) | 1.4 | (1.3 to 1.4) | <0.001 | 1.2 | (1.2 to 1.3) | <0.001 |
| 15–44 | 4661 | 2459 | (52.8) | 1.0 | (1.0 to 1.1) | 0.139 | 1.0 | (1.0 to 1.1) | 0.366 |
| 45–64 | 2789 | 1464 | (52.5) | 1.0 | (1.0 to 1.1) | 0.212 | 1.0 | (1.0 to 1.1) | 0.164 |
| ≥65 | 1169 | 588 | (50.3) | | Ref | | | Ref | |
| Gender | | | | | | | | | |
| Female | 5827 | 3375 | (57.9) | | Ref | | | Ref | |
| Male | 5319 | 2859 | (53.8) | 1.1 | (1.0 to 1.1) | <0.001 | 1.1 | (1.0 to 1.1) | 0.001 |
| Source of referral | | | | | | | | | |
| ASHA | 8428 | 4690 | (55.7) | 1.3 | (1.2 to 1.4) | <0.001 | 1.2 | (1.1 to 1.3) | <0.001 |
| AWW | 88 | 52 | (59.1) | 1.3 | (1.1 to 1.6) | 0.003 | 1.2 | (1.0 to 1.5) | 0.025 |
| RMP | 617 | 271 | (43.9) | | Ref | | | Ref | |
| Community | 2013 | 1221 | (60.7) | 1.4 | (1.3 to 1.5) | <0.001 | 1.3 | (1.2 to 1.4) | <0.001 |
| Previous history of anti-TB treatment | | | | | | | | | |
| Yes | 2501 | 1034 | (41.3) | 0.7 | (0.6 to 0.7) | <0.001 | 0.7 | (0.7 to 0.8) | <0.001 |
| No | 8645 | 5200 | (60.2) | | Ref | | | | |
| Presence of signs and symptoms† | | | | | | | | | |
| Haemoptysis in last 6 months | 1346 | 494 | (36.7) | 0.6 | (0.6 to 0.7) | <0.001 | 0.7 | (0.6 to 0.7) | <0.001 |
| Cough ≥2 weeks | 8895 | 4514 | (50.8) | 0.7 | (0.6 to 0.7) | <0.001 | – | | |
| Sputum | 6184 | 2779 | (44.9) | 0.6 | (0.6 to 0.7) | <0.001 | – | | |
| Chest pain in last 1 month | 7061 | 3260 | (46.2) | 0.6 | (0.6 to 0.7) | <0.001 | – | | |
| Fever ≥2 weeks | 8242 | 4289 | (52.0) | 0.8 | (0.8 to 0.8) | <0.001 | – | | |
| Night sweats ≥2 weeks | 3730 | 1724 | (46.2) | 0.8 | (0.7 to 0.8) | <0.001 | – | | |
| Severe weight loss in last 3 months | 8318 | 4254 | (51.1) | 0.7 | (0.7 to 0.8) | <0.001 | – | | |
| Swelling in a lymph node | 2067 | 1517 | (73.4) | 1.4 | (1.4 to 1.5) | <0.001 | – | | |
| Other factors† | | | | | | | | | |
| Alcohol user | 159 | 90 | (56.6) | 1.0 | (0.9 to 1.2) | 0.862 | – | | |
| Tobacco user | 2280 | 1109 | (48.6) | 0.8 | (0.8 to 0.9) | <0.001 | – | | |

Age is missing for 142 presumptive cases and 98 not evaluated cases.
All % are row percentages.
P value ≤0.05 is considered significant.
*While building the model for multivariate Poisson's regression analysis, all the signs and symptoms except haemoptysis were dropped because of high collinearity assessed on the basis of high variance inflation factor. The final variables in the model were selected on the basis of likelihood ratio testing.
†For signs, symptoms and other factors, the absence of that characteristic sign and symptoms was considered as the reference category.
ASHA, accredited social health activist; AWW, Anganwadi worker; RMP, registered medical practitioner; RR, relative risk; TB, tuberculosis.

Enablers

**Category 1: Logistics-related**
- Transport allowance [a] [b] [c]

**Category 2: Health system-related**
- Free services in the PHS [a] [b]
- Knowledge of the PHS procedures [a]
- Incentive to ASHA for TB work is vital [b] [c]
- Previous positive patient experience in PHS [b] [c]

**Category 3: Health provider-related**
- ASHA's assistance in the diagnostic process [a]
- ASHA's accompaniment to the health facility [a] [c]
- ASHA's understanding of the PHS functioning [a]
- Satisfied people referred others to ASHA [b]

Barriers

**Category 1: Logistics-related**
- Unavailability of public transport from patient's village [a] [c]
- No knowledge of the diagnostic center location [a]
- No transport support for the disabled patient [b]

**Category 2: Health system-related**
- Care at PHC was slow and time-taking [a] [b] [c]
- Not all diagnostics were available in the PHC [b]
- High cost of services not available in the PHC [b]
- Difficult to understand PHC system [a] [b]
- PHC remained closed [a]
- Inability to reach PHC in working hours [a] [b]
- Provider was unavailable at the PHC [a]
- Irregular and unpaid ASHA incentives [b]
- Faith in the private sector [b] [c]
- Poor patient experience deterred others as well [b]

**Category 3: Health provider-related**
- Informal providers misguided the patient [b] [c]
- Low morale of the health providers [b]
- Health worker strikes [a]
- Poor health provider behavior at PHC [b] [c]
- ASHA didn't come to the patient's home [a]

**Category 4: Family-related**
- Poor awareness of TB care in PHS [a] [c]
- Family concerned only when symptoms were severe [b]
- Insufficient family support [a]
- No caretaker in the family to help in reaching hospital [a]
- Family permission was necessary [b]
- Indebted, and no money for healthcare [a]
- Restricted mobility of women [b]

**Category 5: Patient-related**
- Denial of suffering from TB [a]
- Fear of TB [b]
- Discrimination against diseased patient [b] [c]

**Figure 3** The enablers and barriers in diagnostic evaluation from patients', ASHA' and field coordinators' perspective in an ACF TB programme in Samastipur, India, from January 2018 to December 2018. [a]Responses of patient; [b]responses of ASHA; [c]responses of FC. ACF, active case finding; ASHA, accredited social health activist; PHC, primary health centres; PHS, public health system; TB, tuberculosis.

## Category 1: logistics related

As per patients, public transport was not available in certain locations and some patients mentioned their inability to pay for it even where it was available. They also reported not knowing the location of the diagnostic centre.

> I don't know where the diagnostic center is. From my village, I have to walk for 3–4 km before I find any transport. (65-year-old woman)

The ASHA and FC corroborated that transport and insufficient support for certain patient population like disabled people was a key logistics-related challenge.

> Travel options are also limited for some areas. Either it is not readily available or you have to walk before anything can be found. They don't have the money to reserve an entire shared vehicle. If the patient is disabled or elderly, they face even more problems. (Field coordinator)

## Category 2: health system related

According to patients, the care at PHC was slow and took a lot of time. They said either the PHC remained closed or they did not reach the PHC in working hours. At other times when they reached, the providers were not available at the facility.

> I had to go 2–3 times to the PHC before all the tests were completed and reports arrived. Only then I could get the medicines. (17-year-old female patient)

> We have to run-around a lot in the government hospital. There are huge buildings and it is difficult to figure out what happens where alone. (30-year-old male patient)

The health workers reported unavailability of some diagnostic services in the PHC and the high cost of getting these in the private sector. ASHA's remuneration for providing TB care services was irregular and unpaid. They indicated that poor patient experience at the PHC

also deterred others and that the public put a stronger faith in the private sector.

> Patient doesn't know where the registration desk is, where the doctor sits, where the labs and x-ray are done. We have to go with them to get everything completed quickly. (ASHA)

> I have provided treatment to 20 patients, but I haven't been paid for it. It is only in the recent 1–2 years that there has been some improvement. We are not told properly how to fill the document and what to submit for release of payments. All our papers were taken and nothing happened. Earlier we used to only counsel the patient to go for diagnostics, but now we come with them if we get money for helping in diagnosis. (ASHA)

> If patient come on a given day and their work doesn't get done, they won't come again. Coming again will mean foregoing another day's wage. They will say that ASHA cheated them and nothing happened at the hospital and nothing is available at the PHC ever. If they go to a private clinic, everything is done in one day and they get the medicine… They also prefer private. (ASHA)

### Category 3: health provider related

Patients described health worker strikes in the field and at the PHC as a barrier. One patient also reported that ASHA does not come to their home.

> I took the sputum to the PHC, but nothing was done. PHC was on strike for 15 days. The unabated cough and fever didn't let me sleep. When I felt that I would die, I took an injection from the RMP and drank 5 cough syrups. Only then was there any relief. (65-year-old male patient)

> What will happen after going to the PHC? I go to the hospital and come back with nothing. Even ASHA doesn't come. I am going to lose everything in this hustle and bustle. (35-year-old female patient)

The health workers reported RMPs misguiding the patients, low morale of the workers and poor health provider behaviour at the PHC as barriers.

> RMPs misguide the patients a lot. They give medications and injections without any concern. They are not at risk even if the patient dies. If patient becomes serious and they have made money, they will refer them to hospital. They roam through the villages and people get medicine at their doorstep. Someone who is a laborer, their income for the day is saved. (ASHA)

> When we take the patient to the PHC, lab technician will return the sample and asks to come on another day… Doctor will also not write any medication and don't know the updated guidelines… We have to then listen from our patient. (ASHA)

### Category 4: family related

Patients indicated not knowing the extent of available services in the PHS. Some described severe indebtedness of the family and not having any money for healthcare. According to them, insufficient family support and absence of a caretaker in the family were also barriers.

> I live with my two granddaughters and my son works as a watchman in New Delhi. The money is irregular. If I've no money, I won't go. In November, I didn't have money and couldn't go… For consulting with the RMP, I had to take a loan of ₹150 (~USD 2.2). After a week, I've been able to return only ₹50. (65-year-old female patient)

> Both my kids are alone. My husband has migrated for work. He says that take the kids to a senior doctor in a private hospital. It'll cost at least ₹3000 (~USD 44) and I am waiting for it… There is only my old mother-in-law in the house besides me. How will I leave them alone at home? (Mother of a 6-year-old patient)

Health workers mentioned permission from the head of the family to visit facility, restricted mobility of women in the families as compared with men and family's concern only if symptoms were severe as barriers.

> Patients say that we'll ask the head of the family and come only if they assent. Sometimes they are not sure of the quality of care in the PHS. We also counsel the guardian… This happens more often for women. Men are independent; they don't have to ask the head or the wife… It is also difficult for women. They have to complete household chores and care for the kids before leaving home for the facility. Often, by the time they reach, the facility will close. (ASHA)

> If the symptoms are not severe, then patient takes some medicine from RMP at their home and doesn't want to come for a test. (Field coordinator)

### Category 5: patient related

Patients indicated denial of suffering from TB.

> I don't need any diagnostics. I didn't have TB ever in my family. This is all because of gas. (65-year-old patient)

As per health workers, some patients had a fear of TB. They also reported discrimination by the community against TB patients.

> We don't say TB at first. If we say it, people hesitate. Some are offended that we have said TB is a possible disease for them. We've to tell them gently. One sputum positive patient said that she was hurt that I said she has TB. (ASHA)

> A lot of people prefer to keep their disease status hidden. TB is considered contagious and people don't want neighbors to know of it. Earlier, even patient's utensils and bedding were kept separate. (ASHA)

## DISCUSSION

This is the first such study to estimate the PDLFU in an ACF programme while also investigating the barriers and enablers in accessing diagnostic evaluation. About 44% of the presumptive TB cases could not undergo diagnostic evaluation. Provision of transport allowance for patients, accompaniment and support of ASHA, incentives to ASHA and knowledge of the PHS procedures emerged as key enablers. The major barriers were misinformation and stigma, insufficient family and health provider support, transport despite an allowance, inadequate health services and poor faith in the public sector.

The strengths of the study are as follows. First, we used a mixed-methods design in which the quantitative and qualitative components complement each other. The qualitative component explored hitherto unresearched reasons for the PDLFU and recorded the perspectives of both patients and CHWs. Second, the study was conducted in an ACF programme using routinely collected data, hence, reflecting the field reality.

There are several limitations as well. First, the ACF intervention was an add-on to the RNTCP programme with extra provisions like travel allowance. Since the project removed certain barriers in undergoing diagnostic evaluation, the analysis is not necessarily generalisable to routine care. Second, the CHWs and RMPs were likely filtering patients before they were referred to the programme. Therefore, the true number of people screened would be higher than that reported in the study. Similarly, the proportion who were identified as presumptive TB cases in the study is likely much higher than in the overall community. Third, while the presumptive TB cases receiving a test was known, there was no information on cases reaching the facility but not receiving a test, or people not completing the diagnostic process.

While 96.6% of the presumptive TB cases received a CXR in the mobile van in the ACF project in Myanmar, only 51.4% of the cases referred for sputum microscopy to the health facility followed through.[12] Similarly, in an ACF campaign in India, only 22% referred people made it to the DMC, but about 54% received sputum microscopy with provision of specimen transport.[13] In comparison, 41.1% of the presumptive TB cases in our study underwent diagnostic evaluation, which included 81.6% and 50.6% receiving a CXR and sputum microscopy, respectively, at a health facility. Referral with adequate support like that through ASHA accompaniment and community-oriented TB service delivery like mobile vans can complement each other in diagnostic evaluation. In the case of mobile CXR, immediate availability of a diagnostic test for which people did not have to travel encouraged uptake. Furthermore, specimen transport helped in microbiological testing by moving the sample instead of the patient, but challenges like refrigeration and sample preparation are aplenty.[25] Point-of-care diagnostic tests like the upcoming GeneXpert Omni and Truenat can improve uptake of microbiological testing in such scenarios.[26 27] The Zimbabwe study reported place of residence (rural)

and type of facility (private) as important risk factor for prediagnostic LFU.[28] While we found association of PDLFU with various risk factors like age and previous history of TB, they are of limited public health or clinical importance.

Nonetheless, qualitative findings provide rich insights. The transport allowance, free diagnostic services and ASHA's accompaniment supported the patients. While covering the financial costs of the patient certainly helps, opportunity cost like lost wages also need to be considered.[29 30] In fact, barriers like indebtedness of family can exacerbate the opportunity cost, consequently, restricting the benefits of covering financial cost through transport allowance and free diagnostics.[31] Furthermore, systemic factors like unavailability of transport options limits the effectiveness of such solutions.[32 33] The health system and health provider issues like cumbersome processes in health facilities and poor provider behaviour resonated in our findings.[34] These patient narratives add explanation to people's preference for private sector and informal providers even though affordable services are available in the PHS. People's trust in the provider is a crucial factor in determining their preferred provider, which was also corroborated by patients in our study.[35] The disparity is further accentuated by variability of ASHAs' quality.[36] In our study, a patient complained of their ASHA's lackadaisical attitude and another failed to find the diagnostic centre indicating that their ASHA did not help. This difference between ASHAs was also confirmed by the FCs: not all ASHAs are equally interested or equally capable. This will need attention in programmes involving ASHA to encourage their active engagement.

From patients' perspective, stigma, misinformation and insufficient family support emerged as important barriers. The support to patients is often dependent on their family and its situation. For instance, women whose husband migrated for work have to manage the household with limited finances and find it difficult to make time during the labs' visiting hours. Additional challenges of a patriarchal society like restricted mobility and family elder's permission compound the problem for women.[37] Similarly, older and disabled people may not even have the essential family or systemic support. Furthermore, the misinformation and stigma in the community against TB force the patients to keep the diagnosis to themselves. Although the ASHAs reported that stigmatising practices have improved, a thorough investigation needs to be undertaken to further reduce the negative influence of stigma on TB care and treatment.

We suggest a two-pronged approach to improve the situation: a community-level effort focusing on stigma and patient support and a systemic reform in the health system. A sustained campaign on awareness of the disease along with services available for such patients is a must. As the campaign corrects misinformation, it creates a viable climate to tackle the stigma. Delivering such a campaign effectively will need involvement of all sections of the community, in particular the opinion shapers and

influencers in the region.[38] The campaigns will need contextualisation based on the population's language and culture. Both informal and formal community association, like religious groups or self-help groups, can be a possible vehicle.[39] In fact, such groups along with patient support network can be leveraged to create a support system for patients as well.[40] The second prong hinges on building a system of seamless care and improving quality in the public sector.[41 42] Private sector interventions in TB care have demonstrated seamless care through processes like social franchising for diagnostic tests.[43] Similarly, trusted informal providers deliver services at the patient's doorstep at a convenient time, thus, also reducing the opportunity costs. PHS needs to learn from the private and informal providers to be a competitive source of healthcare services.[44–46] It will require investment in taking healthcare closer to the community through steps like upgrading the CHWs, adequate supportive supervision and instituting health outposts.[47] In addition, quality needs to be a core principle in the health system, both private and public. Involving people in assessing the local health system by social audits and patient feedback and embedding quality in the performance indicator of the health system and health workers can be a starting step.

In conclusion, these findings are relevant to the broader primary health services in addition to the TB programme. While improving care in the PHS is essential to make it competitive, involving the community and considering sociocultural factors will be critical to its success.

**Author affiliations**
[1]Department of Research, Innovators In Health, Patna, Bihar, India
[2]Dr. R.P Centre for Ophthalmic Sciences, All India Institute of Medical Sciences, New Delhi, Delhi, India
[3]Department of Operations, Innovators In Health, Patna, Bihar, India
[4]Department of Community and Family Medicine, All India Institute of Medical Sciences, Bathinda, Punjab, India
[5]Department of Consulting, PHTB Consult, Tilburg, The Netherlands
[6]Department of Research, Centre for Development of Human Initiatives, Jalpaiguri, West Bengal, India

**Acknowledgements** We are grateful for the support of Revised National Tuberculosis Control Program (RNTCP), Government of Bihar, State TB Officer Dr (Major) K N Sahai, Communicable Disease Officer Dr Sree Ram Prasad and District Health Society, Samastipur. We would like to thank Sriram Selvaraju, Anthony D Harries, Dick Menzies and participants of the TB Research Methods course at McGill Summer Institute 2018 for their comments on the protocol and Amol Dongre for his feedback on the qualitative results. The programme could not have been implemented without the tireless efforts of ASHAs, health providers and the team at Innovators In Health. This research was conducted through the Structured Operational Research and Training Initiative (SORT IT), a global partnership led by the Special Programme for Research and Training in Tropical Diseases at the WHO (WHO/TDR). The model is based on a course developed jointly by the International Union Against Tuberculosis and Lung Disease (The Union) and Medécins sans Frontières (MSF/Doctors Without Borders). The specific SORT IT programme which resulted in this publication, was jointly developed and implemented by: The Union South-East Asia Office, New Delhi, India; the Centre for Operational Research, The Union, Paris, France; Department of Preventive and Social Medicine, Jawaharlal Institute of Postgraduate Medical Education and Research, Puducherry, India; Department of Community Medicine and School of Public Health, Postgraduate Institute of Medical Education and Research, Chandigarh, India; Department of Community Medicine, All India Institute of Medical Sciences, Nagpur, India; Dr

Rajendra Prasad Centre for Ophthalmic Sciences, All India Institute of Medical Sciences, New Delhi, India; Department of Community Medicine, Pondicherry Institute of Medical Science, Puducherry, India; Department of Community Medicine, Kalpana Chawla Medical College, Karnal, India; National Centre of Excellence and Advance Research on Anemia Control, All India Institute of Medical Sciences, New Delhi, India; Department of Community Medicine, Sri Manakula Vinayagar Medical College and Hospital, Puducherry, India; Department of Community Medicine, Velammal Medical College Hospital and Research Institute, Madurai, India; Department of Community Medicine, Yenepoya Medical College, Mangalore, India; Karuna Trust, Bangalore, India and National Institute for Research in Tuberculosis, Chennai, India.

**Contributors** Conceptualisation of intervention: TG, DS and MaB; study design: all authors; data collection: TG, DS and RM; data analysis and interpretation: TG, VG, DS, MV, RM and MaB; writing — original draft: TG; writing — review and edit: all authors.

**Funding** This project was supported by the Stop TB Partnership's TB REACH initiative and was funded by the Government of Canada and the Bill & Melinda Gates Foundation. MaB and DS are supported by a Grand Challenges Explorations grant number OPP1190735 and OPP1190905, respectively, from the Bill & Melinda Gates Foundation. The training program, within which this paper was developed, was funded by the Department for International Development, UK.

**Disclaimer** No funders had any role in study design, data collection and analysis, decision to publish, or preparation of the manuscript.

**Competing interests** None declared.

**Patient consent for publication** Not required.

**Ethics approval** Ethical approval was obtained from Institutional Ethics Committee, Emmanuel Hospital Association, New Delhi (Order 201) and the Ethics Advisory Group, International Union Against Tuberculosis and Lung Disease, Paris (EAG number 98/18). The committee waived the informed consent requirement for quantitative study. Informed consent was obtained from participants before the in-depth interviews and focus group discussions.

**Provenance and peer review** Not commissioned; externally peer reviewed.

**Data availability statement** Data are available on reasonable request. The deidentified data used in the study arre available from the corresponding author on reasonable request. It contains information on patient demographics and characteristics, screening criteria, diagnostic test used and final diagnosis. There is no additional data available beyond that.

**ORCID iDs**
Tushar Garg http://orcid.org/0000-0002-6781-8574
Vivek Gupta http://orcid.org/0000-0002-6157-3705
Dyuti Sen http://orcid.org/0000-0001-5204-3801
Madhur Verma http://orcid.org/0000-0002-1787-8392
Miranda Brouwer http://orcid.org/0000-0001-8260-4649
Manish Bhardwaj http://orcid.org/0000-0001-7885-8005

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
