## [Reviewer comments · BMJ Open]

ARTICLE DETAILS

TITLE (PROVISIONAL)	Pre-diagnostic loss to follow-up in an active case finding tuberculosis program: a mixed-methods study from rural Bihar, India
AUTHORS	Garg, Tushar; Gupta, Vivek; Sen, Dyuti; Verma, Madhur; Brouwer, Miranda; Mishra, Rajeshwar; Bhardwaj, Manish

VERSION 1 – REVIEW

REVIEWER	Mareli Claassens Stellenbosch University, South Africa
REVIEW RETURNED	15-Sep-2019

GENERAL COMMENTS	Major concerns: More and more papers are showing that asymptomatic individuals who haven't sought care, especially in the community, could form an important proportion of TB cases who are missed. Can the authors comment on the expected number of asymptomatic cases in this context, and would it make sense in a future study to have a different screening algorithm? In line 132, please explain in more detail what it meant to "look for people at risk of TB during their routine work and social activities". Could some presumptive TB cases have been missed? Could selection bias have been introduced? Comment on why the specific diagnostic algorithm was selected. In many countries nowadays, once a patient is symptomatic, GeneXpert is the first diagnostic test. Regarding the adjusted logistic regression model: please comment on the method used to select inclusion of certain determinants. It seems from the results as if all determinants were included independent of their univariable strength of association with the outcome. This has led to a few "strange" p-values, which switched from being non-significant by far, to very significant in the adjusted analysis, and vice versa. Did the authors consider confounding or colaterality? In table 2, it seems some of the reference categories are mixed up. For instance, in the gender category, a higher proportion of females was LFU, but the RR=0.9 with males as reference category. Please check the results carefully. Only seven in-depth interviews were conducted, split between patients who were evaluated and not evaluated (3:4). Do the authors think this number of interviews is sufficient to cover all aspects in detail on why PDLFU is an issue in these communities? Could different blocks and different PHC (and the management thereof) play a role? The authors wrote that patients received transport fees and that diagnostic services were free of charge. Which other costs could
--

	have hampered access of care? Are there any data on catastrophic costs in these communities, for instance? Comment on the clinical relevance of the results: because of the large sample size, many determinants were associated with the outcome. Are these all clinically relevant, or relevant in a public health sense? What could be a practical method to incorporate screening for these determinants in a clinical setting or as part of a regular community-based screening program? Was a waiver of consent approved to include the quantitative data of participants, since only participants of the qualitative aspect gave informed consent? I would like to see more pro-active discussion points on what could be done in communities like these where so many patients are lost from the care cascade pre-diagnostically. Minor concerns: Consider including a flow diagram of the diagnostic algorithm. The authors might mention the quantitative results on gender in the discussion, since it reflects a similar message as the qualitative results (once the quantitative results have been checked). In line 115, "last-mile TB care services" should be explained. There is a typo in line 235: this age group is LESS likely to be LFU. In line 387, "intuitive screening" is mentioned. Could the authors explain this term?
--	--

REVIEWER	Niccolò Riccardi, MD Specialist in Infectious Diseases and Tropical Medicine Infectious Diseases and Tropical Medicine Department IRCCS Ospedale Sacro Cuore Don Calabria, Negrar, Verona Italy
REVIEW RETURNED	14-Oct-2019

GENERAL COMMENTS	The manuscript is an interesting paper about pre-diagnostic loss to follow-up of patients with presumptive symptoms of TB in a rural population in India. I appreciate the idea of the Authors to insert first hand experience (patients, community health workers) in the manuscript to raise awareness of the problems that limit the end of the TB pandemic. I suggest to accept the manuscript after a minor revision. INTRODUCTION 1) In the introduction section, the Authors correctly state that active TB case-finding is necessary to increase early diagnosis. I would suggest them to add that: "Active case finding may increase linkage to care through enhanced TB awareness within the population and decreased TB related stigma" and cite: Riccardi N, Alagna R, Motta I, Ferrarese M, Castellotti P, Nicolini LA, Diaw MM, Ndiaye M, Cirillo D, Codecasa L, Besozzi G. Towards ending TB: civil community engagement in a rural area of Senegal: results, challenges and future proposal. Infect Dis (Lond). 2019 May;51(5):392-394. and Adejumo AO, Azuogu B, Okorie O, Lawal OM, Onazi OJ, Gidado M, Daniel OJ, Okeibunor JC, Klinkenberg E, Mitchell EMH. Community referral for presumptive TB in Nigeria: a comparison of four models of active case finding. BMC Public Health. 2016 Feb 23;16:177.
---

	2) A key role in early diagnose of TB in rural areas is played by molecular diagnostic tool (e.g. GeneXpert MTB/RIF) which decrease the chances of missing a positive case and are less operator-dependent than AFB-stain. I kindly suggest the Author to discuss the role of molecular diagnostic tool in the introduction. GENERAL SETTING Anganwadi workers (AWW) are community health workers are paid to find TB cases: are they paid the same amount of money regardless the number of presumptive TB patients they bring to screening? Please clarify. Moreover, do they receive appropriate training to detect patients with TB symptoms? Please clarify. SPECIFIC SETTING Please explain which kind of follow-up was provided to the patients diagnosed with TB. Were transportation for fu visits paid to the patients? RESULTS The Authors state “one-fifth of the presumptive TB cases reported a previous history of anti-TB treatment”: have they finished previous treatments? It may be interesting to know if they auto-referred for presumptive TB thanks to increased awareness of the diseases.
--	---

REVIEWER	Charlotte Colvin USAID, Virginia
REVIEW RETURNED	25-Oct-2019

GENERAL COMMENTS	This study has a lot to offer in terms of documenting prediagnostic loss to follow up in the context of an active case detection project. The analysis of the cascade is helpful for understanding factors associated with loss to follow up. Unfortunately, the statistical analysis (or at least the presentation of results) is incomplete and confusing. There are only two tables with basic descriptive and bivariate analysis (although Table 2 is not labeled clearly, it is possible that these results are from the multivariate analysis) and the narrative mentions a couple of results from the multivariate analysis (ex, age related risk of LTFU). So one issue that needs to be addressed is the lack of detail on the bivariate vs multivariate analysis - the narrative should be very clear which analysis it refers to when reporting the results, and the tables should be clearly labeled. Additionally, only 7 program beneficiaries were interviewed for this study, yet 27 providers/facilitators were interviewed. If the objective is to better understand the barriers faced by individuals at each step of the cascade, why were so few beneficiaries interviewed? It also seems that issues of stigma were minimized in the discussion in favor of health systems factors, even though there is presentation of results indicating that stigma and misinformation about TB is a problem from the patient's perspective. There are also conclusions in the discussion for which there is no data in the results - for example, the authors say that "word-of-mouth recommendation of satisfied patients" emerges as a key enabler, but in the results from the patients do not mention this is a factor. So a closer review of the findings from the patients themselves to align with the conclusions would be helpful.
---

REVIEWER	Daria Szkwarko Warren Alpert Medical School of Brown University, Department of Family Medicine, Providence, Rhode Island, USA
REVIEW RETURNED	27-Oct-2019

GENERAL COMMENTS	Overall, this is an interesting paper on an important topic – community-based active case finding for TB in a high TB burden community in India. Active, systematic TB screening is an important WHO recommendation that is difficult to implement in many high burden settings. This manuscript focuses specifically on losses in the care cascade between presumptive TB identification and TB evaluation at the facility and reveals perspectives from patients and healthcare workers regarding enablers and barriers related to presumptive TB evaluation. The qualitative results in particular would certainly add to the literature and provide guidance to others globally who are trying to implement active case finding. There are several major changes that I would recommend:  1) The authors describe an evaluation of a TB Reach funded active-case finding program. It is not entirely clear on page 7 which interventions were funded by TB Reach and which are a part of routine RNTCP care. It would be helpful to know if transport allowance provided to presumptive TB patients, ASHA assistance to the patient to reach the centres, and free diagnostic testing were all funded by TB Reach. If all of these interventions were funded by TB Reach, the factor analysis is not necessarily generalizable to routine presumptive TB care in this population since certain barriers have been removed by this project. This is an important limitation that needs to be added to the discussion. 2) It would be great if the authors could please define 'presumptive TB case' within the text and not just in the figure. Did this definition differ for people living with HIV vs not? Was a standard RNTCP definition used or was it defined for this TB Reach project? Was the same definition used for children < 15 years of age? 3) I see that patients who had a diagnostic test after 30 days were excluded from the diagnostic evaluation category and were considered PDLFU. Yet, patients who had a diagnostic test prior to community referral, were counted to have a diagnostic evaluation. How was this decided? Did the authors do an additional factor analysis to confirm that if the patients who had a diagnostic test after 30 days were not included as PDLFU, the factor analysis results would remain the same? 4) Quantitative analysis: 22.4% of presumptive TB cases reported a previous history of anti-TB treatment. Was this a factor that was considered in the analysis? I would be curious to know if those with a previous history of being treated were less likely to come in for an evaluation. Additionally, it is not clear how the signs and symptoms analysis was performed. I think that this will potentially be clearer once the presumptive TB definition is better explained. 5) The qualitative part of this manuscript is robust and impressive and seems more valuable to me than the quantitative analysis. It provides important insight into active-case finding that is not often available in the literature. However, the key points from the qualitative results could be better summarized in the discussion for readers. If space is an issue, I would encourage the authors to consider splitting the evaluations into two manuscripts and focus on the qualitative results. For example, one of the interventions was for the ASHA to help patients find a diagnostic center. Yet in lines 277-278, a patient reports not knowing where the diagnostic center is. Why is that? Are ASHAs unable to help all patients even though this
--

	project encouraged them to do so? Another example are the health provider-related barriers starting in line 311. It is important for the authors to remember that most readers might not have a basic understanding of the healthcare system in India. It would be helpful to add a sentence in the background regarding the public vs private sector as it relates to presumptive TB case evaluation as well as expand upon this in the discussion. 6) The abstract should be strengthened to further describe the design and better summarize the outcomes and results. Please review the abstract for minor errors such as add an 's' in participants in line 37 and add a period at the end of line 48. 7) The conclusions in the abstract are excellent regarding health system strengthening and patient-centered care yet the authors do not provide detailed recommendations based on their findings on how to operationalize this in their discussion in the manuscript. For example, in lines 19-20 the authors recommend strong patient support across the TB care pathway and involvement of families in TB care. How? Were any lessons learned from the qualitative interviews and focus groups that might provide potential solutions? The last sentence of the discussion suggests sputum transportation and adequate support from someone knowing the health system. Yet, patients still experienced difficulty despite ASHA support – how might adequate support be further improved? Specific health strengthening recommendations based on the authors' experiences and qualitative results would greatly strengthen this paper and would make it much more useful to others in the field. Other comments 1) Please include one sentence to summarize TB incidence in India in the introduction – authors include 'a million' for patients not notified but not all readers will be familiar with the burden of TB disease in India. 2) In line 72, authors include stats on South African care cascade PDLFU – this could potentially be removed since this study only focuses on India or moved to lines 83 where other programs are discussed. 3) In the specific setting section, please include one sentence describing what ACF was occurring in this community prior to the TB Reach funded project. Were ASHA's doing any ACF or just conducting treatment support? If none, it would be very helpful to state "Prior to this project, no ACF activities were taking place in the region" to quickly help the reader contextualize the magnitude of the work the authors and team did. 4) In line 73, please add pre-diagnostic and change the abbreviation to PDLFU as this abbreviation is used throughout the rest of the manuscript. 5) Please ensure that all references include a url if available and access date (e.g. line 33, reference 1) 6) Please revise reference spacing in lines 524 and beyond. 7) Consider adding references and incorporating information from: WHO health system strengthening building blocks as well as WHO integrated people-centered care framework to further support conclusions to strengthen aspects of the health system and move towards patient-centered models.
--	---

VERSION 1 – AUTHOR RESPONSE

Reviewer: 1

Reviewer Name: Mareli Claassens

Institution and Country: Stellenbosch University, South Africa

Please state any competing interests or state 'None declared': None declared.

Major concerns:

More and more papers are showing that asymptomatic individuals who haven't sought care, especially in the community, could form an important proportion of TB cases who are missed. Can the authors comment on the expected number of asymptomatic cases in this context, and would it make sense in a future study to have a different screening algorithm?

The number of asymptomatic cases is beyond the scope of our study. In fact, the CHW-led referral system prevents us from estimating the true value of people screened, which would be higher than those referred to the program. Nonetheless, it is an important consideration in the design of programs relying on the symptom-based screening. A community-wide screening with a tool like chest X-ray will be an appropriate tool to ascertain it.

In line 132, please explain in more detail what it meant to "look for people at risk of TB during their routine work and social activities". Could some presumptive TB cases have been missed? Could selection bias have been introduced?

We've added a clarification (line 126). It is a possibility that some presumptive cases were missed because the CHWs likely undertook a level of screening before referring the patients to the program as noted in line 394.

Comment on why the specific diagnostic algorithm was selected. In many countries nowadays, once a patient is symptomatic, GeneXpert is the first diagnostic test.

We used a diagnostic algorithm consistent with the national TB program. Also in India, the national TB program is moving towards using GeneXpert as first test. However, at the time of the study, this was available at a very small scale to our patients.

Regarding the adjusted logistic regression model: please comment on the method used to select inclusion of certain determinants. It seems from the results as if all determinants were included independent of their univariable strength of association with the outcome. This has led to a few "strange" p-values, which switched from being non-significant by far, to very significant in the adjusted analysis, and vice versa. Did the authors consider confounding or colaterality?

Thank you for the observations. We've used a Poisson regression model instead of a logistic model.

Following your suggestion, we have examined collinearity using variance inflation factors (VIF). Collinearity was observed among the symptoms, as well as between tobacco and alcohol use. Therefore, among symptom, we only included history of haemoptysis since it had the highest influence. It is also clinically relevant since patients with haemoptysis are likely to be more alarmed.

We have also used likelihood ratio testing to generate a parsimonious model and have omitted the variables that were not contributing to the model. These details have also been added as a footnote in the table 2, as well as in the data analysis section.

In table 2, it seems some of the reference categories are mixed up. For instance, in the gender category, a higher proportion of females was LFU, but the RR=0.9 with males as reference category. Please check the results carefully.

We have revised the table 2. Please see the above response for details.

Only seven in-depth interviews were conducted, split between patients who were evaluated and not evaluated (3:4). Do the authors think this number of interviews is sufficient to cover all aspects in detail on why PDLFU is an issue in these communities? Could different blocks and different PHC (and the management thereof) play a role?

Our in-depth interview participants came from all the 3 blocks included in the study and sought care in the respective PHCs. Their responses emerging from care in different geographies have been

included in the results. We stopped at seven interviews as no new themes emerged then and we had established saturation of findings.

The authors wrote that patients received transport fees and that diagnostic services were free of charge. Which other costs could have hampered access of care? Are there any data on catastrophic costs in these communities, for instance?

In addition to the financial costs, access to patients may be limited by opportunity costs like the daily wage for people working in the informal sector (line 415). While evidence on catastrophic costs is not available from this particular community, there is evidence from elsewhere in India which mentions the above costs.

Comment on the clinical relevance of the results: because of the large sample size, many determinants were associated with the outcome. Are these all clinically relevant, or relevant in a public health sense? What could be a practical method to incorporate screening for these determinants in a clinical setting or as part of a regular community-based screening program? As we discuss in line 411, the public health value of determinants' association with outcome is minimal because of the large sample size and relatively small difference in absolute numbers. It should also be noted that these were routine programmatic conditions and not a rigorous trial.

Was a waiver of consent approved to include the quantitative data of participants, since only participants of the qualitative aspect gave informed consent?

Yes, waiver of consent was approved by the ethics committee. We've added the details in the section on ethics approval (line 511).

I would like to see more pro-active discussion points on what could be done in communities like these where so many patients are lost from the care cascade pre-diagnostically.

We've added it in the discussion (line 442).

Minor concerns:

Consider including a flow diagram of the diagnostic algorithm.

Since the pre-diagnostic stage includes only the symptomatic screening, we decided to omit the diagnostic algorithm in favour of manuscript's readability.

The authors might mention the quantitative results on gender in the discussion, since it reflects a similar message as the qualitative results (once the quantitative results have been checked).

We've added it in the discussion (line 435).

In line 115, "last-mile TB care services" should be explained.

We've added an explanation in line 106.

There is a typo in line 235: this age group is LESS likely to be LFU.

We've revised the results and discussion in line with changes to table 2.

In line 387, "intuitive screening" is mentioned. Could the authors explain this term?

We mean that the CHWs were likely screening patients at their level before referring them to the program. We've simplified it in the text (line 392).

Reviewer: 2

Reviewer Name: Niccolò Riccardi, MD Specialist in Infectious Diseases and Tropical Medicine
Institution and Country:

Infectious Diseases and Tropical Medicine Department
IRCCS Ospedale Sacro Cuore Don Calabria, Negrar, Verona
Italy

Please state any competing interests or state 'None declared': None declared

The manuscript is an interesting paper about pre-diagnostic loss to follow-up of patients with presumptive symptoms of TB in a rural population in India. I appreciate the idea of the Authors to insert first-hand experience (patients, community health workers) in the manuscript to raise awareness of the problems that limit the end of the TB pandemic.

I suggest to accept the manuscript after a minor revision.

Introduction

1) In the introduction section, the Authors correctly state that active TB case-finding is necessary to increase early diagnosis.

I would suggest them to add that: "Active case finding may increase linkage to care through enhanced TB awareness within the population and decreased TB related stigma" and cite: Riccardi N, Alagna R, Motta I, Ferrarese M, Castellotti P, Nicolini LA, Diaw MM, Ndiaye M, Cirillo D, Codecasa L, Besozzi G. Towards ending TB: civil community engagement in a rural area of Senegal: results, challenges and future proposal. *Infect Dis (Lond)*. 2019 May;51(5):392-394. And Adejumo AO, Azuogu B, Okorie O, Lawal OM, Onazi OJ, Gidado M, Daniel OJ, Okeibunor JC, Klinkenberg E, Mitchell EMH. Community referral for presumptive TB in Nigeria: a comparison of four models of active case finding. *BMC Public Health*. 2016 Feb 23;16:177.

Thank you for the suggestion. We've added the second reference.

2) A key role in early diagnose of TB in rural areas is played by molecular diagnostic tool (e.g. GeneXpert MTB/RIF) which decrease the chances of missing a positive case and are less operator-dependent than AFB-stain. I kindly suggest the Author to discuss the role of molecular diagnostic tool in the introduction.

We appreciate the suggestion. Since the manuscript concerns with aspects before diagnosis, we've not ventured into diagnostics. Nonetheless, we do discuss sputum transport and availability of diagnostic tests.

General Setting

Anganwadi workers (AWW) are community health workers are paid to find TB cases: are they paid the same amount of money regardless the number of presumptive TB patients they bring to screening? Please clarify.

Yes, they are paid the same incentive. We've added a clarification (line 142).

Moreover, do they receive appropriate training to detect patients with TB symptoms? Please clarify.

Yes, they participate in similar trainings organised by the project as the other CHW.

Specific Setting

Please explain which kind of follow-up was provided to the patients diagnosed with TB. Were transportation for fu visits paid to the patients?

Follow-up services were consistent with those provided during the diagnosis, for example, travel allowance for follow-up visits. In addition, the CHW made home visits to provide follow-up support, like, checking for adverse effects (line 139).

Results

The Authors state "one-fifth of the presumptive TB cases reported a previous history of anti-TB treatment": have they finished previous treatments? It may be interesting to know if they auto-referred for presumptive TB thanks to increased awareness of the diseases

While these patients qualify in the WHO's definition of previous history of ATT as per its systematic screening guidelines, they may not necessarily have finished the entire treatment previously.

While it would be pertinent to identify the self-referrals, we do not have means to do it. We only know who referred the person to the program, which in most cases is a CHW. We wouldn't know if a patient self-referred themselves to a CHW.

Reviewer: 3

Reviewer Name: Charlotte Colvin

Institution and Country: USAID, Virginia

Please state any competing interests or state 'None declared': None declared

Please leave your comments for the authors below

This study has a lot to offer in terms of documenting prediagnostic loss to follow up in the context of an active case detection project. The analysis of the cascade is helpful for understanding factors associated with loss to follow up.

Unfortunately, the statistical analysis (or at least the presentation of results) is incomplete and confusing. There are only two tables with basic descriptive and bivariate analysis (although Table 2 is not labeled clearly, it is possible that these results are from the multivariate analysis) and the narrative mentions a couple of results from the multivariate analysis (ex, age related risk of LTFU). So one issue that needs to be addressed is the lack of detail on the bivariate vs multivariate analysis - the narrative should be very clear which analysis it refers to when reporting the results, and the tables should be clearly labeled.

Thank you for your suggestions. We have revised the data analysis section in the methodology. In the revised analysis, we use a Poisson regression model.

Furthermore, based on feedback of the first reviewer, we have examined collinearity using variance inflation factors (VIF). Collinearity was observed among the symptoms, as well as between tobacco and alcohol use. Therefore, among symptom, we only included history of haemoptysis since it had the highest influence. It is also clinically relevant since patients with haemoptysis are likely to be more alarmed.

We have also used likelihood ratio testing to generate a parsimonious model and have omitted the variables that were not contributing to the model. These details have also been added as a footnote in the table 2, as well as in the data analysis section.

We have also labelled the tables clearly. We hope that the results are clearer and interpretable now.

Additionally, only 7 program beneficiaries were interviewed for this study, yet 27 providers/facilitators were interviewed. If the objective is to better understand the barriers faced by individuals at each step of the cascade, why were so few beneficiaries interviewed?

The qualitative findings are a result of 7 in-depth interviews and 3 focus group discussions (involving 27 people). Two FGDs were with CHWs and one was with program staff. While the FGDs resulted in understanding health system factors, they also revealed community-level factors—like stigma—which were not immediately apparent in the patient interviews. We stopped at seven interviews as no new themes emerged then and we had established saturation of findings.

It also seems that issues of stigma were minimized in the discussion in favor of health systems factors, even though there is presentation of results indicating that stigma and misinformation about TB is a problem from the patient's perspective.

We've expanded the discussion on stigma (line 431).

There are also conclusions in the discussion for which there is no data in the results - for example, the authors say that "word-of-mouth recommendation of satisfied patients" emerges as a key enabler, but in the results from the patients do not mention this is a factor. So a closer review of the findings from the patients themselves to align with the conclusions would be helpful.

We've revised the narrative to better align the findings and the conclusion.

Reviewer: 4

Reviewer Name: Daria Szkwarko

Institution and Country: Warren Alpert Medical School of Brown University, Department of Family Medicine, Providence, Rhode Island, USA

Please state any competing interests or state 'None declared': None declared

Comment

Overall, this is an interesting paper on an important topic – community-based active case finding for TB in a high TB burden community in India. Active, systematic TB screening is an important WHO recommendation that is difficult to implement in many high burden settings. This manuscript focuses specifically on losses in the care cascade between presumptive TB identification and TB evaluation at the facility and reveals perspectives from patients and healthcare workers regarding enablers and barriers related to presumptive TB evaluation. The qualitative results in particular would certainly add to the literature and provide guidance to others globally who are trying to implement active case finding.

Major changes

There are several major changes that I would recommend:

- 1) The authors describe an evaluation of a TB Reach funded active-case finding program. It is not entirely clear on page 7 which interventions were funded by TB Reach and which are a part of routine RNTCP care. It would be helpful to know if transport allowance provided to presumptive TB patients, ASHA assistance to the patient to reach the centres, and free diagnostic testing were all funded by TB Reach. If all of these interventions were funded by TB Reach, the factor analysis is not necessarily generalizable to routine presumptive TB care in this population since certain barriers have been removed by this project. This is an important limitation that needs to be added to the discussion.

Indeed, the provisions mentioned above were funded by TB Reach as a part of the intervention. We agree with you and have added this to the discussion (line 390).

- 2) It would be great if the authors could please define 'presumptive TB case' within the text and not just in the figure. Did this definition differ for people living with HIV vs not? Was a standard RNTCP definition used or was it defined for this TB Reach project? Was the same definition used for children < 15 years of age?

We've added the definition in the text as well (line 129). We used the standard RNTCP definition, and it didn't differ for either PLHIV or children less than 15 years of age.

- 3) I see that patients who had a diagnostic test after 30 days were excluded from the diagnostic evaluation category and were considered PDLFU. Yet, patients who had a diagnostic test prior to community referral, were counted to have a diagnostic evaluation. How was this decided? Did the authors do an additional factor analysis to confirm that if the patients who had a diagnostic test after 30 days were not included as PDLFU, the factor analysis results would remain the same?

Thank you for your observation. We have presented the sensitivity analysis in supplementary table 1 at the end of the manuscript with an added explanation in methods as well (line 198). Its results have no effect on our original findings.

- 4) Quantitative analysis: 22.4% of presumptive TB cases reported a previous history of anti-TB treatment. Was this a factor that was considered in the analysis? I would be curious to know if those with a previous history of being treated were less likely to come in for an evaluation. Additionally, it is not clear how the signs and symptoms analysis was performed. I think that this will potentially be clearer once the presumptive TB definition is better explained.

Thank you for your observation. Indeed, previous history of ATT was considered in the analysis and the results are presented in table 2. Please also note that for signs and symptoms and other factors, the absence of that characteristic considered as the reference category. We've not included those rows in the table to ensure brevity. Nonetheless, we have added additional information in the footnote of table 2 to enhance clarity. In addition, the data analysis section and table 2 have been extensively revised.

The case definition of presumptive TB has been mentioned in line 129.

- 5) The qualitative part of this manuscript is robust and impressive and seems more valuable to me than the quantitative analysis. It provides important insight into active-case finding that is not often available in the literature. However, the key points from the qualitative results could be better summarized in the discussion for readers. If space is an issue, I would encourage the authors to consider splitting the evaluations into two manuscripts and focus on the qualitative results. For example, one of the interventions was for the ASHA to help patients find a diagnostic center. Yet in lines 277-278, a patient reports not knowing where the diagnostic center is. Why is that? Are ASHAs unable to help all patients even though this project encouraged them to do so? Another example are the health provider-related barriers starting in line 311. It is important for the authors to remember that most readers might not have a basic understanding of the healthcare system in India. It would be helpful to add a sentence in the background regarding the public vs private sector as it relates to presumptive TB case evaluation as well as expand upon this in the discussion.

We agree that qualitative aspect of the study is more valuable, particularly from a programmatic perspective. However, we think the qualitative discussion benefits from the quantitative analysis, hence, a combined manuscript. We've revised the discussion to include plausible variability in the quality of care by different ASHAs (line 425). We've also added background on health system in India (line 99).

- 6) The abstract should be strengthened to further describe the design and better summarize the outcomes and results. Please review the abstract for minor errors such as add an 's' in participants in line 37 and add a period at the end of line 48.

We've revised the abstract.

- 7) The conclusions in the abstract are excellent regarding health system strengthening and patient-centered care yet the authors do not provide detailed recommendations based on their findings on how to operationalize this in their discussion in the manuscript. For example, in lines 19-20 the authors recommend strong patient support across the TB care pathway and involvement of families in TB care. How? Were any lessons learned from the qualitative interviews and focus groups that might provide potential solutions? The last sentence of the discussion suggests sputum transportation and adequate support from someone knowing the health system. Yet, patients still experienced difficulty despite ASHA support – how might adequate support be further improved? Specific health strengthening recommendations based on the authors' experiences and qualitative results would greatly strengthen this paper and would make it much more useful to others in the field.

We've added this in the discussion (line 442).

Other comments

- 1) Please include one sentence to summarize TB incidence in India in the introduction – authors include 'a million' for patients not notified but not all readers will be familiar with the burden of TB disease in India.
- We've included it in line 68.
- 2) In line 72, authors include stats on South African care cascade PDLFU – this could potentially be removed since this study only focuses on India or moved to lines 83 where other programs are discussed.
- We've removed it.
- 3) In the specific setting section, please include one sentence describing what ACF was occurring in this community prior to the TB Reach funded project. Were ASHA's doing any ACF or just conducting treatment support? If none, it would be very helpful to state "Prior to this project, no ACF activities were taking place in the region" to quickly help the reader contextualize the magnitude of the work the authors and team did.
- We've included it in line 121.
- 4) In line 73, please add pre-diagnostic and change the abbreviation to PDLFU as this abbreviation is used throughout the rest of the manuscript.

We defined PDLFU in relation to presumptive TB cases in the screening campaign. However, the meaning of test-access in the cascade is in relation to the incident cases. We keep it the same to avoid confusion.

- 5) Please ensure that all references include a url if available and access date (e.g. line 33, reference 1)
 We've added the missing information.
- 6) Please revise reference spacing in lines 524 and beyond.
 We've revised the spacing.
- 7) Consider adding references and incorporating information from: WHO health system strengthening building blocks as well as WHO integrated people-centered care framework to further support conclusions to strengthen aspects of the health system and move towards patient-centered models.
 Thank you for the suggestion. We've added additional information on strengthening health system in discussion section (line 442 onwards).

VERSION 2 – REVIEW

REVIEWER	Charlotte Colvin United States Agency for International Development, USA
REVIEW RETURNED	23-Dec-2019

GENERAL COMMENTS	My original comments have been addressed with additional detail in the manuscript.
--

REVIEWER	Daria Szkwarko Warren Alpert Medical School of Brown University, Department of Family Medicine, Providence, Rhode Island, USA
REVIEW RETURNED	21-Dec-2019

GENERAL COMMENTS	This is an excellent revision and all of my major comments and suggestions were well addressed. I only have a few additional minor comments. 1) Line 265: I believe layperson should be laypersons. 2) Line 267: Tense change. Consider changing to 'they were well-placed' 3) Line 352: I do not understand what a 'patient-wise folder' is. Please clarify. 4) Line 395: Change 'in case' to 'in the event that' and add 'a' between had and diagnostic. 5) Line 640-642: Does this also mean that the proportion who were identified as presumptive is likely much higher than the reality? If so, please also include that here. 6) Line 792: Please remove 'to enable' if appropriate.
--

VERSION 2 – AUTHOR RESPONSE

Reviewer 1: More and more papers are showing that asymptomatic individuals who haven't sought care, especially in the community, could form an important proportion of TB cases who are missed. Can the authors comment on the expected number of asymptomatic cases in this context, and would it make sense in a future study to have a different screening algorithm?

Author: The number of asymptomatic cases is beyond the scope of our study. In fact, the CHW-led referral system prevents us from estimating the true value of people screened, which would be higher than those referred to the program. Nonetheless, it is an important consideration in the design of

programs relying on the symptom-based screening. A community-wide screening with a tool like chest X-ray will be an appropriate tool to ascertain it.

The key idea in our response is mentioned with the study's limitations and a further clarification has been added in this revision. Please see lines 396 to 399.

Reviewer 1: Comment on why the specific diagnostic algorithm was selected. In many countries nowadays, once a patient is symptomatic, GeneXpert is the first diagnostic test.

Author: We used a diagnostic algorithm consistent with the national TB program. Also in India, the national TB program is moving towards using GeneXpert as first test. However, at the time of the study, this was available at a very small scale to our patients.

We've included it in line 136.

Reviewer(s)' Comments to Author:

Reviewer: 4

Reviewer Name: Daria Szkwarko

Institution and Country: Warren Alpert Medical School of Brown University, Department of Family Medicine, Providence, Rhode Island, USA

Please state any competing interests or state 'None declared': None declared

Please leave your comments for the authors below:

This is an excellent revision and all of my major comments and suggestions were well addressed. I only have a few additional minor comments.

- 1) Line 265: I believe layperson should be laypersons.
We've corrected it to laypersons (line 128).
- 2) Line 267: Tense change. Consider changing to 'they were well-placed'
We've changed the tense (line 130).
- 3) Line 352: I do not understand what a 'patient-wise folder' is. Please clarify.
We meant separate folders for each patient. We've revised it for clarity in line 153.
- 4) Line 395: Change 'in case' to 'in the event that' and add 'a' between had and diagnostic.
We've included your suggestion (line 195).
- 5) Line 640-642: Does this also mean that the proportion who were identified as presumptive is likely much higher than the reality? If so, please also include that here.
Indeed, it is likely to be higher. We've included it in line 398.
- 6) Line 792: Please remove 'to enable' if appropriate.
We've removed it (line 445).

Reviewer: 3

Reviewer Name: Charlotte Colvin

Institution and Country: United States Agency for International Development, USA

Please state any competing interests or state 'None declared': None declared

Please leave your comments for the authors below

My original comments have been addressed with additional detail in the manuscript.